# Fertility Intention and Influencing Factors for Having a Second Child among Floating Women of Childbearing Age

**DOI:** 10.3390/ijerph192416531

**Published:** 2022-12-09

**Authors:** Yan Xiong, Guojin Jiao, Jiaming Zheng, Jian Gao, Yaqing Xue, Buwei Tian, Jingmin Cheng

**Affiliations:** 1School of Public Health, Shanxi Medical University, Taiyuan 030001, China; 2School of Management, Shanxi Medical University, Taiyuan 030001, China; 3School of Public Health, Southern Medical University, Guangzhou 510515, China; 4School of Health Management, Southern Medical University, Guangzhou 510515, China

**Keywords:** universal two-child policy, fertility intention, floating women, influencing factors, reproductive health

## Abstract

In recent years, an increasing number of women participate in population mobility and most of them are of childbearing age. With the continuous expansion of the population size of this group, their fertility intention will have a great impact on the development of China’s population. Therefore, the aim of this study was to evaluate the fertility intention and influencing factors on having a second child in floating women. This study employed the data from the 2018 National Migrants Dynamic Monitoring Survey data. A self-designed questionnaire was used to collect information, such as socio-demographics and fertility intention. Descriptive statistical analysis was carried out to obtain the basic characteristics of the main variables. Chi-square and ANOVA tests were used to analyze the differences in the basic characteristics between three groups of women (with intention, without intention and unsure about having a second child). Multinomial logistic regression was employed to analyze influencing factors associated with fertility intention among the floating women. The results of this study indicated that only 13.07% of the floating women had the intention to have a second child, while 67.73% had no intention of having another child. In the multivariate analysis, age, gender and age of the first child, reproductive health education, employment status and medical insurance were found to be significant influencing factors of fertility intention (*p* < 0.05), while education level and household registration type were not associated with the desire to have a second child (*p* > 0.05). Overall, after the implementation of the universal two-child policy, floating women of childbearing age have reduced intention to have a second child. Reproductive health education and medical insurance play an important role in ensuring the fertility of floating women. This reminds government departments to consider the above factors comprehensively when formulating the next work plan.

## 1. Introduction

Over the past few decades, countries and regions around the world have faced the challenge of rapid changes in population structure. Specifically, population is changing from youth type to middle-aged type. Trends in population size and age structure are shaped mostly by the levels of fertility and mortality, which have declined almost universally around the globe. The direct result of the reduction of mortality brought about by the progress of medical technology is the increased number of elderly individuals; however, the consequences of the decline in the fertility rate should not be overlooked. According to the World Population Prospects 2022 document issued by the United Nations, the average global fertility rate stood at 2.3 births per woman in 2021, falling from about five births per woman in the mid-twentieth century. Declining fertility rates pose economic, social, cultural and political challenges worldwide. For example, with the education rate of the new generation and the popularization of contraceptives, Spain’s fertility rate will drop to about 1.2 by 2100, which could jeopardize the sustainability of welfare-state systems [1]. Additionally, population resources are closely related to economic development and a study in Japan revealed that low fertility rates has led to Japan’s labor shortage, which could jeopardize the economic lifeline of the entire country [2]. The change in the population structure is bound to affect the process of sustainable development in China—the most populous developing country in the world. Therefore, it is of great significance to understand the reproductive willingness of women of childbearing age and its related factors, in order to cope with the change in population structure.

In China, according to the data from the Seventh National Census, the total fertility rate reached a low of 1.3 in 2020, already below the warning line of 1.5 [3,4], which is closely related to the previous population policy. In the late 1970s, in order to relieve the population pressure and eradicate severe poverty, the one-child policy was implemented in China [5]. During this period, each couple was allowed to bear only one child, which has effectively relieved population pressure. However, some new issues have emerged, such as accelerated population aging, skewed sex ratio and decline in the working-age population [6,7]. To improve the population structure and actively respond to the aging population, the Chinese government has gradually changed its fertility policy, from a selective two-child policy to a comprehensive two-child policy. Although these policies have opened up a new era in the field of population and family planning, they may not quickly change people’s reproductive concepts after more than 30 years of low fertility rate and birth control [8], i.e., as reported by Liu et al., no obvious increase in the fertility rate was brought about by the universal two-child policy [9]. Indeed, population policy is not the only important factor that affects fertility and some studies have suggested that social and economic factors are becoming more important than policy factors [10,11]. This provides a new perspective for the relevant research.

Recently, research on fertility intention as a predictor of fertility behavior [12] has become the focus of fertility discussion. Previous studies have indicated that both individual-level and community-level factors such as age, marital status, education, gender of the first child and residence are associated with fertility intention [13,14,15]. A study in Germany has shown that both delayed marriage and increased number of working women are related to the declined fertility rate [16]. Wang and Learn also revealed that a high level of medical insurance and endowment insurance can improve people’s fertility intention [17]. In addition, some scholars have paid attention to the fertility intentions of special groups, such as AIDS patients, adolescents and cancer patients [18,19,20]. Yet there have been limited data and research on fertility intentions among floating women.

Since the reform and opening up, with the transformation of the economic system and the rapid development of urbanization, China has witnessed a great number of people migrating from rural areas to urban areas [21,22]. Some scholars point out that China’s floating population is an economically oriented flow and the working age population aged 15–64 accounts for the majority [23]. It can be seen that most of the population under this mobility mode is at the childbearing age. It is reported that the floating population in China was about 247 million in 2020, of which 50% are women and the vast majority are women of childbearing age [24,25,26]. With the continuous expansion of the population size of this group, their fertility intention has a great impact on the development of China’s population to a large extent. However, most of the current studies about floating women focus on the following aspects: labor participation, social integration, employment quality, social support, etc. A few scholars have included fertility status as an influencing factor of employment in their research, while few studies have paid attention to the fertility intention of floating women.

Although some results have been confirmed from the previous research on fertility intention, it is unclear whether these results are applicable to floating women. Based on the above, this study focuses on floating women and aims to investigate their fertility intention to have a second child and related influencing factors; the research results have important guiding significance for relevant departments in improving the social security system for floating women.

## 2. Materials and Methods

### 2.1. Study Design and Study Participants

This study used data from the China Migrants Dynamic Survey (CMDS) in 2018, which was an annual national sample survey of the internal migrants organized by the National Health Commission (NHC). The purpose of the CMDS was to understand the changing landscape of internal migration, the utilization of public health services and the management of family planning services [27]. The survey was conducted in 32 provincial units, which covered all 31 provinces and the Xinjiang Production and Construction Corps (XPCC) of China. The distributions of households surveyed across the above 32 provincial units ranged from 2000 to 10,000, including eight sampling units that consisted of 2000, 3000, 4000, 5000, 6000, 7000, 8000 and 10,000 households.

Sample populations were selected by using a stratified multi-stage sampling method with a probability-proportional-to-size (PPS) approach. First, 31 provinces (autonomous regions and municipalities) and XPCCs were taken as the first-level sample units. Then, city units in each province were divided into two tiers. One tier included the mandatory cities, such as the provincial capitals (i.e., Nanjing) and specific major cities (i.e., Suzhou). The second tier included all other cities. Third, in each selected city according to the administrative division, township (town, street) attributes were sorted as the third layer. Next, we selected townships (towns and streets) by the PPS method and the village (neighborhood) committee was selected by the same method as for the selected townships. All eligible subjects in the selected village (neighborhood) committees were invited to participate in the study. Survey participants included the floating population comprising individuals who had lived in the destination for more than one month, who were aged 15 years and older and who were not registered in the district. A household needs to investigate only one floating population.

Face-to-face interviews were conducted by the interviewers trained by local health bureau staff. The survey information covered the basic characteristics of the participants and their family members, as well as employment, basic public healthcare and family planning policy services. Finally, a total of 152,000 floating individuals were surveyed. Among them, 47.7% were male and 52.3% were female.

### 2.2. Dependent Variable

According to the research needs of this study, the dependent variable was the fertility intention for having a second child among floating women of childbearing age. This variable was measured by the following question: “Do you intend to have another child in the future?”, with the response options being “intend”, “unsure” and “do not intend”. We excluded all inapplicable data and finally obtained 24,896 participants, all of whom were women of childbearing age (aged 15–49 years). The specific flowchart of the respondents is shown in Figure 1.

### 2.3. Independent Variables

Demographic characteristics included age, ethnic group (Han or minority), Hukou type (government certificate of legal residency), gender of the first child and age of the first child. Socio-economic characteristics included education level (junior high school or below and senior high school or above) and employment status (employee with employers, employee without employers, employer, self-supporting laborers and others). Migration characteristics included migration scope (cross-province, within-province, within-city) and length of migration.

The utilization of public health services was reflected by social medical insurance and reproductive health education. Social medical insurance was measured by the question: “Which type of social medical insurance have you participated in?”. The five choices were the following: cooperative medical insurance for urban and rural residents (CMIURR), new rural cooperative medical scheme (NRCMS), urban resident based basic medical insurance (URBMI), urban employee-based basic medical insurance (UEBMI) and free medical treatment (FMT). Health education is an important part of basic public health services. Considering that the topic of this study was fertility intention, reproductive health education was included in the study. Specifically, the reproductive health education was assessed by this question: “Have you received reproductive health education in your local community in the past year?” Possible answer was “yes” or “no”.

### 2.4. Data Analysis

Data were processed and analyzed using STATA version 16.0. First, frequencies and percentages were used to describe the floating women’s characteristic variables. Second, Chi-square and ANOVA tests were used to analyze the differences in basic characteristics between the three groups of women (with intention, without intention and unsure about intention to have a second child). Third, to further explore potential influencing factors associated with the floating women’s fertility intention, multinomial logistic regression was used and odds ratios (ORs) and 95% CIs were calculated. All tests were two-tailed and statistical significance was set at the 5% level.

## 3. Results

### 3.1. Socio-Demographic Characteristics

Overall, 24,896 floating women of childbearing age were included in the current study. Table 1 shows the demographic characteristics of all participants. The mean age of the participants was 33.67 ± 7.21 years. Nearly half of them completed the nine-year compulsory education and 49.91% had an education level of high school or above. Regarding Hukou, 64.97% of the floating population had agricultural Hukou. Next, 63.17% of the individuals had a work status of “employee,” including employees with and without employers and 94.61% had medical insurance. In terms of the characteristics of migration, the majority of participants (46.69%) moved across province and the average time in the inflow area was 5.70 years, with a standard deviation of 5.11 years.

### 3.2. Fertility Intention among Floating Women of Childbearing Age

Table 2 summarizes the intention of the floating women of childbearing age who already have one child to have a second child. Among the participants in this study, only 13.07% (3245/24,896) expressed a desire to have a second child, while 67.73% (16,863/24,896) had no intention of having another child and another 19.20% (4779/24,896) had not yet decided. Univariate analysis showed that the fertility intention among floating women was significantly associated with migration range and duration, age, education level, ethnicity, hukou, gender of the first child, age of the first child, reproductive health education, employment status and medical insurance (*p* < 0.05).

### 3.3. Influencing Factors of Fertility Intention

To explore the influencing factors on fertility intention among the floating women, multinomial logistic regression was carried out. Socio-demographic variables were defined as independent variable X and fertility intention was defined as dependent variable Y, as shown in Table 3. The findings indicated that age (*OR*_1_ = 1.110, *OR*_2_ = 1.043), gender of the first child (*OR*_1_ = 0.601, *OR*_2_ = 0.693), age of the first child (*OR*_1_ = 1.027, *OR*_2_ = 0.958), reproductive health education (*OR*_1_ = 1.287), medical insurance (*OR*_1_ = 1.420) and migration range (*OR*_1_ = 0.842) and duration (*OR*_2_ = 0.977) were the main factors influencing fertility intention among the floating women. However, education level and household registration type were not associated with the floating women’s desire to have a second child (*p* > 0.05).

## 4. Discussion

As the country with the largest elderly population in the world, the data from the Seventh National Population Census in China showed that the population aged 60 years and above accounts for 18.70%. Based on this fact, the Chinese government has successively issued the “universal second child” policy and the three-child policy [28] (that is, a couple can have three children and enjoy supporting measures) to address the serious challenges of population aging. However, factors such as high housing prices, high child-rearing costs and imperfect maternity services inhibit the fertility willingness of people of childbearing age, especially those in low-income groups [4]. The declining fertility rates over the past decade present the possibility that current childbearing cohorts will have fewer children than their predecessors [29]. For the floating population, due to the particularity of their working and living environment, this phenomenon seems more obvious.

In the current study, we found that the second-child intention rate among the floating women was 13.07%, which is much lower than that reported by other researchers. For example, Liu et al. conducted a cross-sectional study in 11,911 Chinese women of childbearing age and found that, after the universal two-child policy, the second-child intent rate was 39.4% in central and eastern China [30]. A hospital-based cross-sectional study in Chongqing showed that, among 814 nulliparous pregnant women, 51.2% intended to have a second child [31]. Although the differences between these studies may be related to the different characteristics of the study population, it is also evident that the floating women have relatively low fertility intention. This is related to the particularity of the floating women. With the development of social economy, the gender structure of the floating population has changed and an increasing number of women participate in population mobility. According to the national dynamic monitoring survey of floating population in 2016, more than 70% of women float for the purpose of seeking greater economic benefits. Considering this reason, it seems reasonable that floating women do not want to have a second child.

According to relevant data, the majority of floating women in China are in the age range of 20–39 years, with low educational level, mainly junior high school, and most of them are married, which was confirmed in this study. Moreover, their fertility intention also deserves more attention. Our study showed that age was one of the most important factors associated with the second-child intention, which is consistent with previous research results [14,32].

A large number of studies have shown that, considering women’s physical and mental health, the best childbearing age for women is before the age of 35 [33]. This is because women bear the main responsibility for pregnancy and birth, while older women are more likely to experience infertility and pregnancy complications [34,35]. Thus, with the increase of age, women are more likely to choose not to have a second child. This also suggests that primary medical institutions should give full play to their responsibilities and provide full cycle, all-round and multi angle reproductive health services for women of childbearing age in different life periods.

This study showed that the sex of the first child significantly influenced the floating women’ intention to have a second child. In the current study, 60% of the floating women’s first child was a boy, which seems worth thinking about. Influenced by traditional culture, male preference has a long history in China. Yet, with the development of the economy, women’s education level and employment participation rate have been increasing and the concept of equality between men and women has been widely advocated in the new era. In addition, in order to alleviate the imbalance of the sex ratio between men and women, China’s government has also promulgated corresponding laws to prohibit unlawful fetal gender authentication and selective birth. Against this background, sex-selective abortion has been effectively controlled. Therefore, further research is needed to clarify the reasons behind this result. Compared with the floating women whose first child was a boy, women whose first child was a girl were more likely to have a second child. This may be related to the following reasons. Similar to other countries with Asian culture, China has historically preferred male to female offspring [6]. This is because it is generally believed that male children are more valuable and more likely to support their parents in old age; hence, some parents chose to have a son over a daughter [36]. Although the concept of equality between men and women is widely advocated in the new era, the influence of traditional ideas has not been completely eliminated. Therefore, when the first child is a girl, individuals are more willing to have a second child. Obviously, this phenomenon also exists in the floating population. An aim of the implementation of the universal two-child policy is also conductive to promoting the balance of the sex ratio at birth to a certain extent.

Reproductive health (RH) care is defined as the constellation of methods, techniques and services that contribute to RH and well-being by preventing and solving sexual health problems [37]. The achievement of universal access to RH care services for all individuals has been emphasized and accepted worldwide [38]. Specifically, RH services include health consulting, contraceptive and birth control services, prenatal and postnatal care services, breast examinations and cervical cancer screenings, etc. In order to achieve universal access to basic public health services, the National Plan for Basic Public Health Services was implemented in 2009 in China, including reproductive health education. Community health service institutions carry out health education and publicity activities in five forms (e.g., materials for health education and knowledge, lectures) every year, including reproductive health education. Therefore, generally speaking, community residents can receive at least one form of reproductive health education every year. We observed in this study that floating women who had not received RH education were less likely to want to have a second child. This is related to the insufficient utilization of public health services by the floating population in the inflow area. Previous studies have shown that migrant women use less RH services than non-migrant women, in terms of prenatal care, hospital deliveries, postpartum visits and system administration [39,40]. Due to mobility, women may not be familiar with the RH services in the inflow area. In addition, the upbringing, care and education of children are time and energy consuming. The busy work schedule, coupled with the parenting of multiple children, can exhaust floating women [8]. These factors may lead to the floating women’s unwillingness to have children again.

For a long time, economic cost has been undoubtedly considered an important factor affecting fertility behavior. According to Liu et al. [30], 47.7% of Chinese women of childbearing age reported economic barriers as the main obstacle to having a second child. As a part of the social security system, medical insurance is an important approach to insuring the basic medical rights of individuals and the economy against loss. Therefore, medical insurance was included in the analysis as an influencing factor. We found that medical insurance was helpful in improving the floating women’s desire to have a second child. Medical insurance, especially maternity insurance, can effectively reduce the cost and risk of having a second child and protect women’s reproductive rights [41,42]. This means that increasing the participation rate of floating women in medical insurance, especially in maternity insurance, could help to improve their desire to have a second child.

In recent years, how women balance the relationship between family and work has been a topic of academic concern. Kahn et al. were among the first to note that a conflict between work and non-work roles is a major source of stress [43]. Indeed, for many mothers, “simultaneously raising children and pursuing a career is challenging because both pursuits are time-consuming” [44]. Therefore, after the implementation of the universal two-child policy, some women tend not to have a second child because of their pursuit of professional value. In this study, we found that floating women with different employment status made different choices for childbearing. Compared with employees, employers and self-supporting laborers had a higher fertility desire, which was related to their more flexible employment time. Moreover, they did not have to worry about unemployment due to pregnancy. This also reminds us that a flexible employment market is of great significance in encouraging women to have children.

According to previous studies, scholars have different views on the relationship between educational level and fertility intention. Some studies have found that women with higher education levels tend to have less desire to have more children [32,45,46]. This is because women with a higher education level tend to have better employment opportunities and women with jobs are less likely to desire more children. However, other scholars have found that education level and labor participation rate significantly positively correlate with fertility intention [47,48]. However, none of the above results seems to be applicable to floating women. Thus, longitudinal design and qualitative research can be considered to further analyze the relationship between educational level and fertility intention among the floating women.

## 5. Limitations

There are several limitations in this study that should be addressed. First, the data source for this study was the China’s Floating Population Health Plan Dynamics Monitoring Survey from 2018. Due to data limitation, the information collected in this survey was more relevant to the personal characteristics of the floating women. Yet fertility intention is often affected by multifaceted factors, not only by individual and economic factors, but also by family relations and social support [49,50]. Therefore, these factors also need to be included in research. Second, this cross-sectional survey provided valuable preliminary insights into fertility intention among floating women with one child but did not allow for the determination of causal relationships. Further studies, such as longitudinal studies and qualitative research, are needed to clarify the subject of the floating women’s fertility intention, so as to provide reference for the government in formulating more appropriate policies.

## 6. Conclusions

Under the general trend of global fertility decline, China has also joined the low-fertility-rate countries. Our study showed that, under the universal two-child policy, the overall intention to have a second child among the floating women was low, as only 13.07% had the desire to have additional children. Age, gender and age of the first child, RH education, employment status and medical insurance were found to be significant influencing factors of the floating women’s fertility intention. The findings of this study have a certain reference value for further optimizing fertility policy and promoting the long-term balanced development of the population. Based on the above-mentioned factors, policy makers should develop and implement targeted policies and measures to ensure that floating women have less worries when deciding to give birth.

## Figures and Tables

**Figure 1 ijerph-19-16531-f001:**
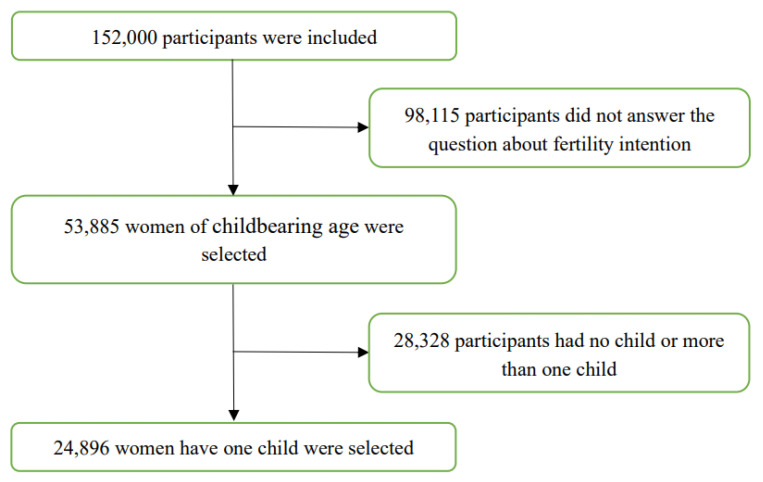
Flowchart of participant selection.

**Table 1 ijerph-19-16531-t001:** Characteristics of the participants (*N* = 24,896).

Variables	*n*	%
Migration range		
Cross-province	11,624	46.69
Within-province	8713	35.00
Within-city	4559	18.31
Time in inflow area (years)	5.70 ± 5.11	
Age	33.67 ± 7.21	
Education level		
Middle school or below	12,470	50.09
High school or above	12,426	49.91
Ethnicity		
Han	23,156	93.01
Minorities	1740	6.99
Hukou		
Agricultural	16,176	64.97
Non-agricultural	4147	16.66
Others	4573	18.37
Gender of the first child		
Boy	14,960	60.09
Girl	9936	39.91
Age of the first child (years)	8.54 ± 7.15	
Reproductive health education		
Yes	13,484	54.16
No	11,412	45.84
Employment status		
Employee with employer	11,013	59.36
Employee without employer	707	3.81
Employer	1314	7.08
Self-supporting laborer	5340	28.78
Others	179	0.97
Medical insurance		
Yes	23,359	94.61
No	1330	5.39

Missing data: Employment status = 6343; Medical insurance = 207.

**Table 2 ijerph-19-16531-t002:** Fertility intention to have a second child among participants.

Variables	Intention for Second Childbirth			*F*/*χ*^2^
	Yes (*n* = 3254)	No (*n* = 16,863)	Unsure (*n* = 4779)	
Migration range				37.21 ***
Cross-province	1450 (44.56)	8059 (47.79)	2115 (44.26)	
Within-province	1226 (37.68)	5693 (33.76)	1794 (37.54)	
Within-city	578 (17.76)	3111 (18.45)	870 (18.20)	
Migration duration	4.94 ± 4.10	6.17 ± 5.48	4.56 ± 3.98	230.57 ***
Age	30.20 ± 4.63	35.26 ± 7.57	30.46 ± 5.19	1400.84 ***
Education level				119.74 ***
Middle school or below	1491 (45.82)	8848 (52.47)	2131 (44.59)	
High school or above	1763 (54.24)	8015 (47.53)	2648 (55.41)	
Ethnicity				33.60 ***
Han	2963 (91.06)	15,787 (93.62)	4406 (92.20)	
Minorities	291 (8.94)	1076 (6.38)	373 (7.80)	
Hukou				43.74 ***
Agricultural	2170 (66.69)	10,778 (63.92)	3228 (67.55)	
Non-agricultural	465 (14.29)	2981 (17.68)	701 (14.67)	
Others	619 (19.02)	3104 (18.41)	850 (17.79)	
Gender of the first child				278.89 ***
Boys	1577 (48.46)	10,674 (63.30)	2709 (56.69)	
Girls	1677 (51.54)	6189 (36.70)	2070 (43.31)	
Age of the first child	5.71 ± 3.74	9.97 ± 7.78	5.44 ± 4.58	1135.20 ***
Reproductive health education				131.59 ***
Yes	1953 (60.02)	8713 (51.67)	2818 (58.97)	
No	1301 (39.98)	8150 (48.33)	1981 (41.03)	
Employment status				33.45 ***
Employee with employer	1309 (57.89)	7591 (59.22)	2113 (60.82)	
Employee without employer	56 (2.48)	538 (4.20)	113 (3.25)	
Employer	183 (8.09)	916 (7.15)	215 (6.19)	
Self-supporting laborers	690 (30.52)	3659 (28.55)	991 (28.53)	
Others	23 (1.02)	114 (0.89)	42 (1.21)	
Medical insurance				22.92 ***
Yes	3087 (95.57)	15,757 (94.14)	4515 (95.64)	
No	143 (4.43)	981 (5.86)	206 (4.36)	

Note: *** *p* < 0.001.

**Table 3 ijerph-19-16531-t003:** Logistic regression analysis of the influencing factors on fertility intention.

Variables	Intention for Second Childbirth	
	No	Unsure
Migration range		
Cross-province	1	1
Within-province	0.842 (0.757–0.937)	0.974 (0.862–1.101)
Within-city	1.000 (0.874–1.146)	1.048 (0.898–1.223)
Migration duration	0.990 (0.979–1.001)	0.977 (0.964–0.990)
Age	1.110 (1.094–1.126)	1.043 (1.026–1.060)
Education level		
Middle school or below	1	1
High school or above	0.970 (0.870–1.082)	0.931 (0.822–1.055)
Ethnicity		
Han	1	1
Minorities	0.729 (0.614–0.865)	0.826 (0.681–1.003)
Hukou		
Agricultural	1	1
Non-agricultural	1.115 (0.969–1.284)	1.000 (0.851–1.176)
Others	0.977 (0.864–1.104)	0.914 (0.794–1.052)
Gender of the first child		
Boys	1	1
Girls	0.601 (0.547–0.660)	0.693 (0.622–0.772)
Age of the first child	1.027 (1.010–1.043)	0.958 (0.941–0.976)
Reproductive health education		
Yes	1	1
No	1.287 (1.169–1.416)	1.069 (0.957–1.193)
Employment status		
Employee with employer	1	1
Employee without employer	1.170 (0.866-1.583)	1.271 (0.906–1.783)
Employer	0.735 (0.614–0.880)	0.746 (0.603–0.923)
Self-supporting laborers	0.744 (0.667–0.830)	0.892 (0.788–1.009)
Others	0.754 (0.471–1.207)	1.046 (0.625–1.753)
Medical insurance		
Yes	1	1
No	1.420 (1.094–1.843)	1.047 (0.773–1.419)

Missing data: Employment status = 6343; Medical insurance = 207.

## Data Availability

The data used in this paper were provided by the National Health Commission of the People’s Republic of China, which is the top agency governing health issues in China. The data, itself, is third party data and the authors did not produce any of the original data.

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
