# Peer review of "Fertility Intention and Influencing Factors for Having a Second Child among Floating Women of Childbearing Age"

_ijerph, 2022, doi:10.3390/ijerph192416531_

Round 1

Reviewer 1 Report

I think fertility intention is a very important issue in public health field. There exist many studies investigating this topic. However, the authors should clarify the research gap between previous related works and this work. As a result, I cannot figure out the motivation of this work. In the discussion part, the authors should highlight the findings in this work, rather than just list findings in previous studies. I cannot find the research contribution of this work after reading it.

Author Response

Dear reviewer:

We are very grateful to your careful reading, thoughtful comments and valuable suggestions for improving the quality of our manuscript (ijerph-2045649) for a possible publication in International Journal of Environment Research and Public Health. We have carefully evaluated these comments, responded to these suggestions point-by-point, and have incorporated modifications into the original manuscript where appropriate. 

Reviewer 1:

I think fertility intention is a very important issue in public health field. There exist many studies investigating this topic. However, the authors should clarify the research gap between previous related works and this work. As a result, I cannot figure out the motivation of this work. In the discussion part, the authors should highlight the findings in this work, rather than just list findings in previous studies. I cannot find the research contribution of this work after reading it.

Response: Thanks for your valuable suggestion, it is very meaningful for us to improve our manuscript. As you said, fertility intention is a very important issue in public health field, and China is no exception. In order to improve the population structure and actively respond to the aging population, the Chinese government is also gradually changing its previous fertility policy. After the promulgation of the universal two-child policy, some scholars have investigated the fertility intentions of people of childbearing age. However, few people seem to pay attention to the fertility intention of floating women.

Due to rapid urbanization processes, China has witnessed a great number of people migrating from rural to urban areas during the past 30 years, called a “floating population” or “internal migrants” [1-3]. With family migration becoming a new trend in the migration process in China, the number of female floating population is increasing. However, most of the current studies about floating women focus on the following aspects: labor participation, social integration, employment quality, social support, etc. A few scholars have included the fertility status as an influencing factor of employment in their research, while few studies have paid attention to the fertility intention of floating women. Although some results have been confirmed from the previous research on fertility intention, it is unclear whether these results are applicable to floating women. This study just fills this gap, and the results of this study have important guiding significance for relevant departments to improve the social security system of floating women, especially reproductive health services.

In addition, some of the discussions did repeat the previous research results, so we revised the discussion section according to your comments.

  1. Zhu J, Ye Z, Fang Q, Huang L, Zheng X. Surveillance of Parenting Outcomes, Mental Health and Social Support for Primiparous Women among the Rural-to-Urban Floating Population. Healthcare (Basel). 2021;9(11):1516.
  2. Xiong Y, Xue Y, Jiao G, Xie J, Cheng J. Comparative Analysis of the Status and Influencing Factors of Immunization Among Children Between Registered and Floating Population. Front Public Health. 2022;10:872342.
  3. Zhao Y, Kang B, Liu Y, Li Y, Shi G, Shen T, Jiang Y, Zhang M, Zhou M, Wang L. Health insurance coverage and its impact on medical cost: observations from the floating population in China. PLoS One. 2014;9(11):e111555.

Reviewer 2 Report

This is a well-performed analysis of fertility intentions among floating women in China, a large population with important implications for China's birth rate in the future, and highly relevant to population and health policy in the world's most population country. The results are not surprising, but they are important, and are reasonably described. The data appears to be of good quality, collected by a major government agency. The statistical analysis is sound.

p. 3, line 47: How was reproductive health education assessed? Was it a question about school education? A little clarification.

p. 5, Table 1: It is remarkable that 60% of the sample had a first child that was a boy. Do the authors have any explanation for this? Is it the product of sex-selective abortion, or children not reported? Some mention of this is warranted.

p. 7, line 205: "Falling period fertility rates during the last decade present the possibility that current childbearing cohorts will have fewer children than their predecessors." This sentence is copied from this source: www.jstor.org/stable/48681363, page 2043. 

p. 8, line 274: "simultaneously raising children and pursuing a career is challenging because both pursuits are time-consuming." This is a direct quote from the source cited, and needs to be in quotation marks.

Author Response

Dear reviewer:

We are very grateful to your careful reading, thoughtful comments and valuable suggestions for improving the quality of our manuscript (ijerph-2045649) for a possible publication in International Journal of Environment Research and Public Health. We have carefully evaluated these comments, responded to these suggestions point-by-point, and have incorporated modifications into the original manuscript where appropriate. Please see the attachment.

Round 2

Reviewer 1 Report

This manuscript has been improved well, I suggest accepting it.